# The Influence of Material Type and Hardness on the Number of Embedded Abrasive Particles during Airborne-Particle Abrasion

**DOI:** 10.3390/ma15082794

**Published:** 2022-04-11

**Authors:** Beata Smielak, Leszek Klimek

**Affiliations:** 1Department of Dental Prosthetics, Medical University of Lodz, ul. Pomorska 251, 92-213 Lodz, Poland; 2Department of Materials Research, Institute of Materials Science and Engineering, University of Technology, ul. Stefanowskiego 1/15, 90-924 Lodz, Poland; leszek.klimek@p.lodz.pl

**Keywords:** airborne-particle abrasion, Cr/Co alloy, Ni/Cr alloy, Ti, ZrO_2_

## Abstract

(1) Background: This paper aims to determine the influence of hardness on the number of abrasive material grains (SiC) embedded on the surface metal alloys and ZrO_2_ during abrasion. (2) Methods: Cylindrical samples were created: 315 made of Cr/Co, Ni/Cr or Ti, and 315 made of sintered ZrO_2_- 3TPZ-Y. These were divided into four groups (each *n* = 35 samples), and were treated with SiC grain sizes 50, 110, and 250 µm at pressures 0.2, 0.4, or 0.6 MPa. The samples were then observed in SEM to study SE and BSE. The surface coverage of abrasive material particles was determined by quantitative metallography. Five samples from each group were subjected to hardness measurements. The results were compared with three-factor variance analysis with using the post hoc Tukey test. (3) Results: The highest amount (40.06) of embedded abrasive was obtained for Ti alloy with a gradation of 250µm at a pressure of 0.6 MPa. The smallest amount of embedded grain (2.66) was obtained for ZrO_2_ for the same treatment parameters. (4) Conclusions: The amount of embedded abrasive particles depends on the type of treated material, gradation particles, and the amount of applied pressure. Harder treated materials are more resistant to grains of abrasive becoming embedded on surface.

## 1. Introduction

When creating dentures, a suitable surface can be formed by airborne-particle abrasion: such treatment results in material expansion, an increase in the geometrical shape in comparison to the real one, an increase in surface roughness and the formation of craters improving the penetration of liquid ceramic, and alterations in physicochemical properties such as electrostatic potential and free surface energy [1]. Surface expansion increases surface energy per nominal unit of surface area and removes weakly attached overhangs and metal flakes formed during the grinding process; this ensures better anchoring, better binding of the coatings deposited on it, and increases surface wettability [2,3]. It also increases the bonding surface of the materials. Furthermore, airborne-particle abrasion forms a homogeneous uniform surface needed for stronger material bonding [3,4,5,6].

The surface needs to be rough to encourage the formation of mechanical abutments, which can anchor applied and fired ceramic masses. The correct preparation of rough surfaces can also improve stress distribution by increasing the energy dispersion during the breaking strength at the material interface [7]. However, for some metal alloys, increased roughness accelerates the corrosion processes. Examples include stainless steel, copper, or titanium alloys [8,9,10]. Airborne-particle abrasion with zirconium oxide can affect mechanical properties of the compound [11,12]. Excessively aggressive operation can result in unfavourable tetragonal-to-monoclinic phase transformation (tm) [13,14,15]. In addition, during airborne-particle abrasion, abrasive particles with high kinetic energy can become embedded in the treated material, which can lead to contamination of the abraded surface [16,17]; such contamination was found to result in a poorer mechanical anchorage of dental ceramics, reduced corrosion resistance, and deteriorated biocompatibility in a titanium surface [17]. Furthermore, impurities change the topography of the surface by creating a discontinuous structure, which may result in the formation of cracks in the veneering porcelain [18].

Undoubtedly, the presence of particles in the material reduces the smoothness of the surface [17,19]; however, the effect of the embedded particles on fired porcelain is not entirely clear. While they expand the treated surface, which may improve the quality of material interface, they may initiate cracks in the ceramics [18]. Embedded abrasive grains may also react with fired ceramics. Therefore, as the interface between ceramics with zirconium oxide or titanium alloys is the weakest point of prosthetic restorations, and contributes to chipping and fracturing, there is a need to determine the effect of different aspects of production on the numbers of embedded grains.

The aim of the study is to determine the impact of selected parameters of airborne-particle abrasion on the amount of embedded SiC grains on the surface of the following alloys: Cr/Co, Cr/Ni, Ti, and sintered ZrO_2_. It also examines the relationship between the amount of embedded abrasive particles and metal hardness.

## 2. Materials and Methods

A total of 1260 cylindrical samples with a diameter of 9 mm and height of 5 mm were prepared from four materials (*n* = 315 samples each): three groups were made from the metal alloys Cr/Co (Heraenium^®^ P, Heraeus Holding GmbH, Hanau, Germany), Ni/Cr (Wiron 99, BEGO USA Inc., Lincoln, USA), or Ti (Tritan CpTi 1, DENTAURUM GmbH & Co. KG, Ispringen, Germany) while the fourth was made of sintered ZrO2- 3TPZ-Y (Cermill, Amann Girrbach AG, Koblach, Austria). The samples were divided into groups (*n* = 35); these were treated with SiC grain sizes 50, 110, 250 µm at pressures of 0.2, 0.4 or 0.6 MPa (Table 1).

The surface topography of the samples was then observed in a scanning electron microscope (SEM, HITACHI S3000-N, Hitachi, Ltd., Tokyo, 100-8280 Japan); the procedure used secondary electrons (SE) and material contrast with backscattered electron (BSE) light. Ten images were taken at randomly selected locations of each disc. In each case, the same imaging field was used for each sample. In total, 12,600 different calculations regarding the amount of embedded grain were made in randomly selected locations in individual samples. The surface coverage of the abrasive material particles was determined by quantitative metallography using Metillo software. Five samples from each group were subjected to hardness measurements with a KB Prüftechnik hardness tester at a load of 9.81N (1 kG) using the Vickers method.

The disks were first ground on a rotary grinder (Metasinex) with SiC abrasive paper grit size of 220, 400, 600, and 800 under water cooling, to ensure a uniform surface before airborne-particle abrasion. Next, the discs were washed in an ultrasonic washer (Quantrex 90 WT, L&R Manufacturing, Inc., Kearny, NJ, USA) in ethyl alcohol for 10 min and dried with compressed air. Airborne-particle abrasion was conducted using a Mikroblast Duo device (Prodento - Optimed, Warsaw, Poland). The abrasive material was embedded at an angle of 45 degrees at a distance of 10 mm. The abrasion time of the specimens was established at 20 s.

The areas with abrasive grains embedded on the surface were determined by material contrast resulting from the difference in chemical compositions, as confirmed previously [16,20]. Example images of metal alloy and zirconium dioxide samples after airborne-particle abrasion were presented in BSE electrons are shown in Figure 1.

Dark areas indicating differences in the chemical composition were visible on the sample surfaces following airborne-particle abrasion. The surface coverage of the abrasive material particles was determined by quantitative metallography using Metillo software [19]. The procedure consisted of the following steps: Briefly, the microscopic image was loaded into the Metillo software and subjected to the following adjustments: shadow correction, normalization of the grey level histogram, manual binarization of the image, and calculation of the percentage surface share of dark (red) areas, i.e., of abrasive elements embedded in the sample surface. An example image of manual binarization of a ZrO_2_ sample after abrasion with SiC (110 μm particles, 0.2 MPa) is given in Figure 1d. The red areas visible in Figure 2 are SiC particles embedded on the sample surface.

The presented photos were only included to show the differences in the number of particles. It was not possible to take pictures with magnification markers, because they would be treated by the calculator as objects to be analysed and would distort the results of the calculations. All photos were taken at the same 500× magnification. In addition, the presented photos were not obtained directly from a scanning microscope—they were subjected to various procedures to allow the calculating program (MetiIlo) to detect them and obtain the most reliable result.

Statistical analyses were performed using the statistical package PQStat version 1.8.2.218. The results were compared with three-factor variance analysis considering the type of abrasion particle, its gradation and applied pressure followed by the post hoc Tukey’s test. *p* < 0.05 was adopted as significant, and *p* < 0.01 as highly significant.

## 3. Results

Table 2 presents the amount of SiC particles embedded in four different materials according to abrasion parameters. Figure 3 shows a graphical interpretation of the results. Significantly higher levels of embedded abrasive (40.06; *p* < 0.05) were obtained for the Ti alloy, with a gradation of 250 μm at a pressure of 0.6 MPa. The smallest amount of embedded grain (2.66) was obtained for ZrO_2_ abrasion with a gradation of 250 μm at a pressure of 0.6 MPa; however, at this particle size, no significant difference in mean embedded abrasive (*p* > 0.05) was observed compared to samples treated with 0.4 MPa (2.79) or 0.2 MPa (3.08). In general, for ZrO_2_ abrasion with a gradation of 250 μm, irrespective of the value of the applied pressure, the amount of embedded abrasive was small and made up a uniform group (labelled with the letter “a”). Higher numbers of embedded grains were observed for ZrO_2_ abrasion with a gradation of 110 μm or 50 μm. Then, irrespective of the applied pressure, the mean amounts of embedded abrasive formed another uniform group (labelled with the letter “d”). All other combinations of materials and abrasion variables resulted in significantly (*p* < 0.05) higher amounts of embedded abrasive compared to ZrO_2_. Means marked with the same letter indicate no significant difference between them (*p* > 0.05).

Table 3 presents descriptive statistics of the amounts of embedded abrasive particles (SiC) with regard to the studied parameters, as well as a table of analysis of variance. All interactions between the factors were highly significant (*p* < 0.0001). In general, the type of material subjected to airborne-particle abrasion had a highly significant influence (*p* < 0.0001) on the amounts of embedded abrasive. The greatest amount of embedded grain was observed for the Ti alloy and the smallest for ZrO_2_. Highly significant differences (*p* < 0.0001) were also observed between gradations. The lowest level of embedded abrasive was noted for 50 μm grain size, and the highest for 250 μm, indicating that the amount of embedded abrasive is positively correlated with gradation.

Highly significant differences (*p* < 0.0001) were observed between pressure and abrasion, and between different pressure values. The smallest numbers of embedded grains were observed for a pressure of 0.2 MPa, and the greatest at 0.6 MPa. Hence, an increase in pressure appears to be associated with an increase in the amount of embedded abrasive.

The results of the Vickers HV1 hardness testing is included in Table 4 (load: 9.81 N/1 kG).

The greatest hardness was observed for ZrO_2_ (mean 1412 HV) and the smallest for Ti alloy (mean 96 HV), with the hardness for ZrO_2_ being about 15 times greater than for Ti alloy. The mean Ni/Cr alloy was mean 186 HV and Co/Cr was mean 403 HV.

## 4. Discussion

Material contrast observations of the treated samples revealed the presence of abrasive material grains embedded on the surface [16,20]. The following processes may take place during abrasion: grains perform the cutting work and rebound off the surface, grains perform the cutting work and remain embedded in the surface, or grains penetrate the surface without performing the cutting work. In all cases, the grains carried by the compressed air carry with them a certain energy depending on their mass and speed, and this influences the quality of the cutting process, and whether they rebound or are absorbed by the surface. It should be remembered that this will also be influenced by their orientation at the point of contact with the treatment surface: abrasive grains are irregular polygons. Therefore, the type of abrasive grain and type of treated material will both influence the process parameters [20,21].

The quantity of embedded abrasive particles appears to be dependent on the type of treated material, the gradation of the abrasive particles, and the amount of applied processing pressure. The tested materials were characterized by varying hardness from very soft titanium (96 HV) through slightly harder Ni/Cr alloy (186 HV), harder Cr/Co alloy (403 HV) up to very hard ZrO_2_ (1412 HV). It should be noted that the hardness of ZrO_2_ is similar to that of the used abrasives. The greatest amount of embedded abrasive, regardless of the type of abrasive material and processing parameters, was observed on the surfaces of titanium alloy samples, followed by Ni/Cr and Co/Cr alloys, and the lowest for the surfaces of ZrO_2_ samples. Clearly, harder materials demonstrated lower numbers of embedded grains at the same abrasion parameters. This is mainly related to the ductility of the treated material: materials of low hardness are characterized by high ductility and can hence easily be embedded by the abrasive particles. The area involved by the embedded particles can expand many times. Depending on the used parameters, it ranges from 3–4% for ZrO_2_, through 13–25% for Cr/Co alloy, 13–30 for Ni/Cr alloy, up to 22–38% for Ti alloy.

Both the size of the abrasive grain and the pressure are positively correlated with the size of the surface area involved by the impacted particles. However, these relationships are not as pronounced as for the type of material, the differences being only a few per cent.

It seems that a five-fold increase in the grain size from 50µm to 250µm should result in a large difference in the area involved by these grains, assuming that the number of embedded grains is similar. It should be noted, however, that contact between the impacting grain and the treated surface may result in the grain chipping and cracking. Thus, the whole grain is not embedded but only a fragment, which would result in a smaller number of embedded grains than expected. In addition, embedding and fixing larger grains is more difficult and requires more energy. Pressure also appears to have a similar influence on the number of embedded grains.

Embedding abrasive grains into the treated surface of prosthetic materials has serious practical consequences. It can be used to clean metal alloy castings and prepare surfaces before ceramic firing. It more often refers to Ni/Cr and Co/Cr alloys. It should be noted that in the titanium–ceramic system, this is the only way to increase bond strength.

Abrasion is not very effective in the case of zirconium oxide, due to its high hardness; in addition, ZrO_2_ is also resistant to etching, even with HF. Therefore, it is recommended to improve the quality of the veneering interface in the case of zirconium oxide/ceramic bonds [11,22,23]. One promising method for this is laser treatment.

Derand and Herø [24] and Gilbert et al. [17] found that Al_2_O_3_ particles can become embedded up to a depth of 10 μm in treated surfaces. The surface area for bonding to the ceramic can be limited. The use of 250 μm gradation aluminium oxide significantly improved the bond strength of the ceramic to the titanium alloy compared to 50 μm gradation, suggesting that the embedded particles support bonding [24]. However, it may be a result of more favourable surface expansion during abrasion with a larger grit.

Yamada et [25] al and Smielak et al. [16] found particle abrasion to be a very effective method of preparing ceramic–titanium binding sites, with the treatment improving the adhesion of ceramics to titanium twofold compared to ground samples [26].

Co/Cr and Ni/Cr alloys demonstrate higher bond strengths to ceramic compared to titanium, even at the same airborne-particle abrasion parameters. However, this is due to the Co-Cr and Ni-Cr alloys forming chemical bonds with the ceramic resulting from mutual dissolution of oxides on the alloy surface and in the ceramic. Indeed, a two- to threefold difference in the magnitude of the bond strength has been noted between Co/Cr or Ni/Cr alloys with ceramics compared to titanium alloys [24,27,28,29], and the adhesion force of porcelain to titanium alloy was found to be 47–64% of the value of the force for the Cr/Ni-ceramic alloy combination [30,31].

It can be concluded that airborne-particle abrasion has a beneficial effect on the preparation of most surfaces of materials for bonding with other materials. It is also often an essential stage in creating a mechanically strong bond, as the resulting rough surface provides a better anchorage for the substrate material and facilitates the application of coating particles. Increased application pressure will increase the kinetic energy of abrasive grains and thus also increase the intensity of the cutting process [4,5,6], resulting in greater mechanical anchorage and increased surface wettability [3,32]. However, further research is still needed to gain a more precise explanation of the role of embedded abrasive particles.

## 5. Conclusions

After airborne-particle abrasion, embedded abrasive particles remain on the treated surface.The amount of embedded abrasive particles depends on the type of treated material, its gradation, and the amount of applied pressure.Harder treated materials are more resistant to abrasive grains becoming embedded on surface.

## Figures and Tables

**Figure 1 materials-15-02794-f001:**
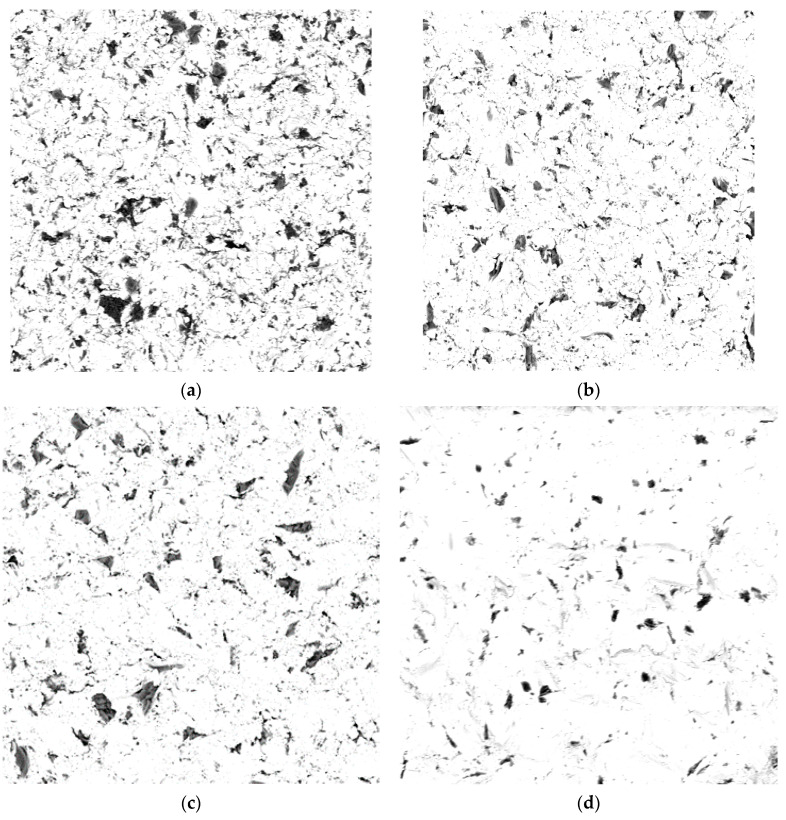
Images of Cr/Co alloy surface after abrasion with SiC 110 μm particles and under pressure of 0.2 MPa BSE obtained with (magnification 500×): (**a**) Ti alloy, (**b**) Ni/Cr alloy, (**c**) Cr/Co alloy, and (**d**) ZrO_2_.

**Figure 2 materials-15-02794-f002:**
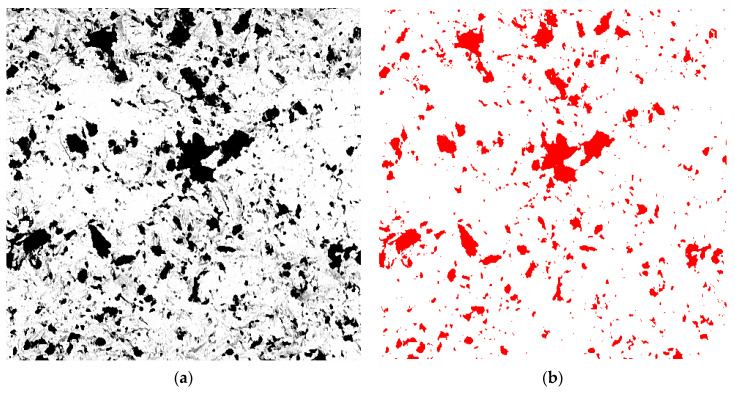
Binarization of microscopic image: (**a**) initial image and (**b**) image after binarization (for calculations).

**Figure 3 materials-15-02794-f003:**
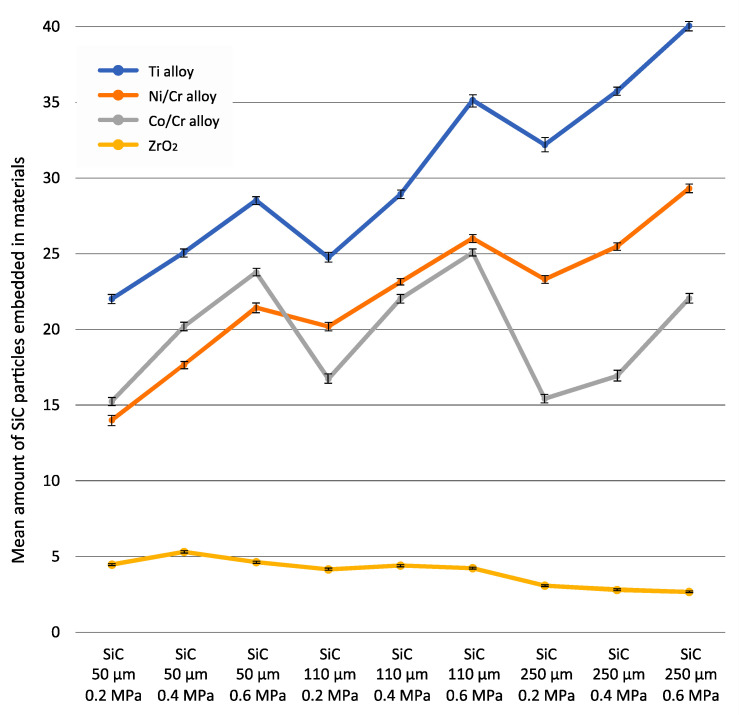
The amount of SiC particles embedded in the four different test materials according to abrasion parameters.

**Table 1 materials-15-02794-t001:** Materials and airborne-particle abrasion parameters.

Material	Gradation (μm)	Pressure (MPa)
alloy Ti	50	0.2 0.4 0.6
	110	0.2 0.4 0.6
	250	0.2 0.4 0.6
alloy Ni/Cr	50	0.2 0.4 0.6
	110	0.2 0.4 0.6
	250	0.2 0.4 0.6
alloy Co/Cr	50	0.2 0.4 0.6
	110	0.2 0.4 0.6
	250	0.2 0.4 0.6
ZrO_2_	50	0.2 0.4 0.6
	110	0.2 0.4 0.6
	250	0.2 0.4 0.6

**Table 2 materials-15-02794-t002:** Descriptive statistics of the amount of SiC embedded abrasive according to material and airborne-particle abrasion parameters.

Material	Gradation (μm)	Pressure (MPa)	Arithmetic Mean	Standard Deviation	Standard Error of the Mean	Uniform Groups (Tukey’s Post Hoc Test)
alloy Ti	50	0.2	22.02	1.78	0.30	jk
		0.4	25.09	1.58	0.27	mn
		0.6	28.53	1.53	0.26	o
	110	0.2	24.76	1.90	0.32	mn
		0.4	28.94	1.68	0.28	o
		0.6	35.15	2.41	0.41	r
	250	0.2	32.19	2.79	0.47	p
		0.4	35.73	1.56	0.26	r
		0.6	40.06	1.83	0.31	s
alloy Ni/Cr	50	0.2	14.01	2.00	0.34	e
		0.4	17.66	1.44	0.24	h
		0.6	21.44	1.90	0.32	ij
	110	0.2	20.20	1.66	0.28	i
		0.4	23.16	1.26	0.21	kl
		0.6	26.01	1.56	0.26	n
	250	0.2	23.31	1.52	0.26	kl
		0.4	25.48	1.48	0.25	n
		0.6	29.31	1.65	0.28	o
alloy Co/Cr	50	0.2	15.25	1.58	0.27	ef
		0.4	20.20	1.66	0.28	i
		0.6	23.77	1.47	0.25	lm
	110	0.2	16.74	1.81	0.31	gh
		0.4	22.03	1.69	0.29	jk
		0.6	25.07	1.34	0.23	mn
	250	0.2	15.43	1.64	0.28	fg
		0.4	16.92	2.05	0.35	h
		0.6	22.05	1.89	0.32	jk
ZrO_2_	50	0.2	4.47	0.40	0.07	cd
		0.4	5.30	0.46	0.08	d
		0.6	4.63	0.38	0.07	d
	110	0.2	4.15	0.41	0.07	cd
		0.4	4.40	0.43	0.07	cd
		0.6	4.23	0.36	0.06	cd
	250	0.2	3.08	0.33	0.06	ac
		0.4	2.79	0.37	0.06	a
		0.6	2.66	0.30	0.05	a

**Table 3 materials-15-02794-t003:** Descriptive statistics of the amount of embedded abrasive particles (SiC) with regard to the studied parameters, as well as a table of analysis of variance.

		Arithmetic Mean	Standard Deviation	Standard Error of the Mean	Uniform Groups (Tukey Post Hoc Test)	
**Material**	Ti	30.28	5.93	0.33	d	
	Ni/Cr	22.29	4.64	0.26	c	
	Co/Cr	19.72	3.90	0.22	b	
	ZrO_2_	3.97	0.94	0.05	a	
**Gradation (μm)**	50	16.86	8.09	0.39	a	
	110	19.57	9.96	0.49	b	
	250	20.75	12.49	0.61	c	
**Pressure (MPa)**	0.2	16.30	8.78	0.43	a	
	0.4	18.97	9.92	0.48	b	
	0.6	21.91	11.71	0.57	c	
		F			*p*	Eta-squared partially
**Material**		16280.70			<0.0001	0.9756
**Gradation (μm)**		709.34			<0.0001	0.5368
**Pressure (MPa)**		1407.50			<0.0001	0.6970
**Material *Gradation**		519.00			<0.0001	0.7178
**Material *Pressure (MPa)**		168.22			<0.0001	0.4519
**Gradation*Pressure (MPa)**		11.30			<0.0001	0.0356
**Material *Gradation*Pressure (MPa)**		8.67			<0.0001	0.0784

**Table 4 materials-15-02794-t004:** HV1 hardness test results of the analysed materials.

Material			
Alloy Ti	Alloy Ni/Cr	Alloy Co/Cr	ZrO_2_
94	187	405	1410
95	186	400	1405
97	186	396	1410
101	184	422	1399
90	189	393	1432
98	182	402	1416
X mean = 96	X mean = 186	X mean = 403	X mean = 1412
SD = 3.8	SD = 2.4	SD = 10.2	SD = 11.3

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
