# Peer review of "The Influence of Material Type and Hardness on the Number of Embedded Abrasive Particles during Airborne-Particle Abrasion"

_materials, 2022, doi:10.3390/ma15082794_

Round 1
Reviewer 1 Report
Dear authors,
This manuscript describes the influence of material type and hardness on the number of embedded abrasive particles during airborne-particle abrasion.
The analysis method is properly described in the revised manuscript and the relationship of result (data set) with the statistics of embedded abrasive particles is well-explained. In my opinion, this manuscript can be accepted in the present form.
Author Response
Comments and Suggestions for Authors
Review 1
Reviewer: This manuscript describes the influence of material type and hardness on the number of embedded abrasive particles during airborne-particle abrasion.
The analysis method is properly described in the revised manuscript and the relationship of result (data set) with the statistics of embedded abrasive particles is well-explained. In my opinion, this manuscript can be accepted in the present form.
Answer: Thank You very much for Your comments.
Reviewer 2 Report
The paper seeks to evaluate the influence of the nature of the material (metallic alloys: Cr/Co, Ni/Cr, Ti and a ZrO2-3TPZ-Y ceramic) in the number of hard particles embedded in the surface of the samples by the action of SIC particles disperses in an airflow. The size of the abrasive (50, 110, 250 μm) and air-jet pressure (0.2, 0.4 or 0.6 mPa) was also varied. The article presents an impressive and even exhaustive statistical significance (1260 samples, 315 of each material). Furthermore, the well-written introduction, in addition to perfectly contextualizing the work, clarifies the envisaged application: the establishment of a homogeneous and uniform surface needed for stronger material bonding in dental prosthetics applications.
Although using relatively trivial techniques (hardness and image analysis) and a somewhat superficial discussion of the results, the paper has a strong point in its statistical significance guaranteeing robustness of the results and conclusions established, thus deserving to be published in Materials. However, while the language is understandable, the text might benefit from some editing by a professional. In addition, the paper presents a few omissions/deficiencies that must be clarified before its final evaluation.
- Even though apparently well established in Dentistry, the designation Airborne-Particle Abrasion sounds strange to a tribologist like me. According to the classic tribological literature (for example Hutchings, I.M and Shipway, P., Tribology: friction and wear of engineering materials, Butterworth-Heinemann- Elsevier, 2017), "in abrasive wear, material is removed or displaced from a surface by hard particles, or sometimes by hard protuberances on a counterface, forced against and moving along the surface" while the "wear caused by hard particles striking the surface, either carried by a gas stream or entrained in a flowing liquid is called erosion." Can the authors comment on that?
- The same observation applies to the widespread use of the word abrasion throughout all text. For me, it would be more appropriate to use erosion.
- The abstract should be a single paragraph and should follow the style of structured abstracts, but without headings. Still, in the abstract, the authors claim that "the quantity of embedded abrasive grains depends on the kind of abrasion particle". As only the SIC was used, this affirmative needs to be modified.
- Since the tribological phenomena are highly dependent on the system, e.g. of the imposed tribological parameters, it is necessary to detail them:
- What is the mass of particles impinged in each testing /
- What is the speed of particles at the initial instant of the impact?
- Why was the 45° impingent angle chosen?
- What would be the probable effect of lower angles? And more perpendicular?
- Please include bars allowing to evaluation the magnification used in the images of Figures 1 and 2
- . Although it is probably correct, using images obtained by back-scattered electrons does not guarantee an accurate determination of the chemical composition (only the presence of light/heavy elements - atomic number). An EDS analysis would give more robustness to the research.
- Please include errors bars in the graphics of Figure 3 in order to show, graphically, that they are (or not) significantly different.
- Please include errors bars in the graphics of Figure 3 in order to show, graphically, that they are (or not) significantly different.
- In my opinion, Table 3 is unnecessary. It would be enough to include the average and standard deviation of the results. In addition, the hardness value of the zirconia reported in the text differs from that shown in the table.
- In discussing the results, the authors make several affirmations, even still plausible, that have a speculative character. Experimental evidence would be very welcome.
Author Response
Review 2
Reviewer: The paper seeks to evaluate the influence of the nature of the material (metallic alloys: Cr/Co, Ni/Cr, Ti and a ZrO2-3TPZ-Y ceramic) in the number of hard particles embedded in the surface of the samples by the action of SIC particles disperses in an airflow. The size of the abrasive (50, 110, 250 μm) and air-jet pressure (0.2, 0.4 or 0.6 mPa) was also varied. The article presents an impressive and even exhaustive statistical significance (1260 samples, 315 of each material). Furthermore, the well-written introduction, in addition to perfectly contextualizing the work, clarifies the envisaged application: the establishment of a homogeneous and uniform surface needed for stronger material bonding in dental prosthetics applications.
Although using relatively trivial techniques (hardness and image analysis) and a somewhat superficial discussion of the results, the paper has a strong point in its statistical significance guaranteeing robustness of the results and conclusions established, thus deserving to be published in Materials. However, while the language is understandable, the text might benefit from some editing by a professional. In addition, the paper presents a few omissions/deficiencies that must be clarified before its final evaluation.
- Even though apparently well established in Dentistry, the designation Airborne-Particle Abrasion sounds strange to a tribologist like me. According to the classic tribological literature (for example Hutchings, I.M and Shipway, P., Tribology: friction and wear of engineering materials, Butterworth-Heinemann- Elsevier, 2017), "in abrasive wear, material is removed or displaced from a surface by hard particles, or sometimes by hard protuberances on a counterface, forced against and moving along the surface" while the "wear caused by hard particles striking the surface, either carried by a gas stream or entrained in a flowing liquid is called erosion." Can the authors comment on that?
Answer: The term may sound strange, but in our opinion, it is perfectly correct. The article does not describe the process of material consumption (e.g. tribological), but the process of surface treatment with the abrasive blasting method. And in this machining, micro-cutting are elementary processes, and are similar to those observed, for example, when grinding with abrasive discs. We mention that in the blasting process, the surface is not impacted, and even if it is, its share is minimal: the process was carried out at an angle of 45 to the surface, so hard particles moved over the surface. As the reviewer noted, during abrasive wear (in our work - processing), the material is removed by hard particles (in our case, abrasive grains) moving along the surface. In our opinion, the term abrasion is equal to abrasive wear.)
Reviewer: The same observation applies to the widespread use of the word abrasion throughout all text. For me, it would be more appropriate to use erosion.
Answer: I think that one answer can be given to both questions.
Reviewer: The abstract should be a single paragraph and should follow the style of structured abstracts, but without headings. Still, in the abstract, the authors claim that "the quantity of embedded abrasive grains depends on the kind of abrasion particle". As only the SIC was used, this affirmative needs to be modified.
Answer: The abstract has been corrected.
Reviewer: Since the tribological phenomena are highly dependent on the system, e.g. of the imposed tribological parameters, it is necessary to detail them:
- What is the mass of particles impinged in each testing?
- What is the speed of particles at the initial instant of the impact?
Answer: We are unable to give the mass of the particles and their velocity. The speed of the grains depends mainly on their mass, which depends on the size of the particles. According to the standards (Federation of European Producers of Abrasives), FEPA 42-D-1984, the stated grain size refers only to the mean size: a mean value of 110 μm can encompass grains in the range 100 - 120 µm. Consequently, the particles have a range of masses and hence different velocities. The mass and speed are also influenced by the shape of the abrasive particles, which can vary considerably (as shown in the photo below). While it may theoretically be possible to calculate an average particle velocity, we believe that this does not make sense. In everyday practice, there is no way to set the particle speed on the abrasive blasting apparatus; the only parameters are the type of abrasive, its size and working pressure, and these are given in the paper).
Reviewer: Why was the 45° impingent angle chosen?
Answer: A dental prosthesis is a multi-faceted element with planes at different angles. As such, it is rotated and inclined at different angles during its processing. Therefore, it can be assumed that the abrasive grains fall on it at various angles from 0⁰ to 90⁰ - the 45⁰ category as assumed as the average value.)
Reviewer: What would be the probable effect of lower angles? And more perpendicular?
Answer: We did not deal with this topic and it was not the aim of the work. However, changing the angle certainly affects the intensity of processing, as well as the number of particles driven into the surface. Regarding the practical aspect of the work, we found that it does not make sense to study the influence of the incidence angle because it varies during processing. Considerations on this subject for titanium are presented in: Gołebiowski M., Wolowiec E., Klimek L .: Airborne-particle abrasion parameters on the quality of titanium-ceramic bonds. J Prosthetic Dentistry 2015, vol 113, issue 5: 453- 459.
Reviewer: Please include bars allowing to evaluation the magnification used in the images of Figures 1 and 2.
Answer: We could include magnification markers; however, we chose not to. The presented photos were only included to show the differences in the number of particles. It was not possible to take pictures with magnification markers, because they would be treated by the calculator as objects to be analyzed and this would distort the results of the calculations. All photos were taken at the same 500x magnification. In addition, the presented photos were not obtained directly from a scanning microscope – they were subjected to various procedures to allow the calculating program (MetiIlo) to detect them and obtain the most reliable result.
It has been added in the article.
Reviewer: Although it is probably correct, using images obtained by back-scattered electrons does not guarantee an accurate determination of the chemical composition (only the presence of light/heavy elements - atomic number). An EDS analysis would give more robustness to the research.
Answer: Of course, the BSE images do not give the exact composition. We relied on our previous research and experience. However, knowing the base material and its composition, its average mass / atomic number of the substance can be calculated. The same can be done with the abrasive. As we know which component is heavy and which is light, we can tell whether the image will be light or dark.)
Reviewer: Please include errors bars in the graphics of Figure 3 in order to show, graphically, that they are (or not) significantly different.
Answer: It has been included.
Reviewer: In my opinion, Table 3 is unnecessary. It would be enough to include the average and standard deviation of the results. In addition, the hardness value of the zirconia reported in the text differs from that shown in the table.
Answer: The results included in Table 3 are part of our study. The hardness value of materials has been corrected in the text.
Reviewer: In discussing the results, the authors make several affirmations, even still plausible, that have a speculative character.
Answer: In discussing the results, we rely on published studies in reputable journals.
Reviewer 3 Report
The presented article does not correspond to the scientific level and there are serious shortcomings in the field of commenting on the results or coherence of experimental ideas. The article is not inserted into the template, which is a serious shortcoming and I also recommend inserting images into the article with higher resolution and definition of individual structures. Fig.2 has no informative value - what does it mean: (for calculations)? The tables in the article are not sufficiently described in the text. Figure 3 shows a graphical interpretation of the results- of any results or specific?
I recommend reworking the article, eliminating visible shortcomings and inserting it into the template.
Author Response
Review 3
Reviewer: The presented article does not correspond to the scientific level and there are serious shortcomings in the field of commenting on the results or coherence of experimental ideas. The article is not inserted into the template, which is a serious shortcoming and I also recommend inserting images into the article with higher resolution and definition of individual structures. Fig.2 has no informative value - what does it mean: (for calculations)? The tables in the article are not sufficiently described in the text. Figure 3 shows a graphical interpretation of the results- of any results or specific?
I recommend reworking the article, eliminating visible shortcomings and inserting it into the template.
Answer: The article has been inserted into the template. "For calculations ", was added to indicate that the number of particles impressed had to be calculated in some way. This was done using the MetiIlo metallographic quantification program. The direct microscopic image is not suitable for this kind of calculation. In order for the program to be able to reliably perform its calculations, the image had to be transformed into a more readable format, i.e. the "computation image".
Reviewer 4 Report
In this work, the impact of selected parameters of abrasion of airborne particles on the amount of SiC grains incorporated into the surface, namely Cr/Co, Cr/Ni, Ti, and sintered ZrO2, was studied. The relationship between the amount of incorporated abrasive particles and the hardness of the metal was also studied.
Congratulations. The work is good, the work is well documented with references/studies by other authors and it can be seen that there is a lot of work, however, there are some concerns about your work, which can be resolved, improving the work and your understanding. Please see the attachment.

Author Response
Review 4
Reviewer: In this work, the impact of selected parameters of abrasion of airborne particles on the amount of SiC grains incorporated into the surface, namely Cr/Co, Cr/Ni, Ti, and sintered ZrO2, was studied. The relationship between the amount of incorporated abrasive particles and the hardness of the metal was also studied.
Congratulations. The work is good, the work is well documented with references/studies by other authors and it can be seen that there is a lot of work, however, there are some concerns about your work, which can be resolved, improving the work and your understanding, such as:
Answer: Thank You very much for Your comments.
Reviewer: The abstract must be in a single text and does not need to have the words “Background”, “Methods”, “Results”, and “Conclusion”. The formatting used by Materials magazine is different from the existing one.
Answer: The abstract has been corrected.
Reviewer: In the abstract, the authors state that “A total of 1260 cylindrical samples comprising four groups of 315 samples: Cr/Co, Ni/Cr, Ti, and sintered ZrO2-3TPZ-Y.”. 1260/4 = 315. That's ok. Then it says that “Each group was divided into four groups (each n=35 samples).” It's confusing. Must be more clear
Answer: The text in abstract has been corrected.
Reviewer: In the "materials and methods", if the authors added a table or a “tree” scheme to explain the creation of the groups, it would be easier to understand. The 4 groups of 315 give a total of 1260 samples. Okay, got it. From here, I couldn't understand the division of the samples, taking into account that n is equal to 35 and taking into account the grain size and pressures.
The number of samples?
35 samples of SIC 50 μm at 0.2 MPa + 35 samples of SIC 50 μm at 0.4 MPa + 35 samples of SIC 50 μm at 0.6 MPa = 105 samples
35 samples of SIC 110 μm at 0.2 MPa + 35 samples of SIC 110 μm at 0.4 MPa + 35 samples of SIC 110 μm at 0.6 MPa = 105 samples
35 samples of SIC 250 μm at 0.2 MPa + 35 samples of SIC 250 μm at 0.4 MPa + 35 samples of SIC 250 μm at 0.6 MPa = 105 samples
Having a total of 315 samples.
Make a table with this information. It is simple and quick to interpret. As shown in table 1, in the results. Suggestion: make a table horizontally (taking the 3 columns turns them into 3 rows).
Answer: According Your suggestion table with this information has been added.
Reviewer: Suggestion: Figures 1 and 2 would look better if they had the scale on the figure itself, as they appear in many works in this area.
Answer: We could include magnification markers; however, we chose not to. The presented photos were only included to show the differences in the number of particles. It was not possible to take pictures with magnification markers, because they would be treated by the calculator as objects to be analyzed and this would distort the results of the calculations. All photos were taken at the same 500x magnification. In addition, the presented photos were not obtained directly from a scanning microscope – they were subjected to various procedures to allow the calculating program (MetiIlo) to detect them and obtain the most reliable result.
Reviewer: The discussion is very rich and grounded. However, the conclusions are very poor. I recommend that in front of each statement, the authors develop a little more, referring to their work and comparing it with works by other authors. It doesn't need to be as extensive as in the discussions but in a summarized form.
Answer: Thank You very much for Your comments about discussion. Conclusions are specific statements resulting from research. Their concise form is clear, and understandable for the reader.
Round 2
Reviewer 3 Report
Článek prošel dostatečnou revizí a v této podobě jej doporučuji zveřejnit v plném znění.
This manuscript is a resubmission of an earlier submission. The following is a list of the peer review reports and author responses from that submission.
Round 1
Reviewer 1 Report
The manuscript submitted is not of the standard expected for this journal. The manuscript should be improved significantly by showing a mathematical foundation and a sound experimental design.
Reviewer 2 Report
The authors studied the effect of the SiC particle sizes and pressure on the amount of SiC abrasive particles embedded on the Cr/Co, Ni/Cr, Ti and ZrO2 surfaces. The amount of SiC particles embedded on the surfaces increased with an increase in the pressure and the particle size, and a reduction in the workpiece. These results were not unexpected. The particles were small and accumulated in various locations of the surface of the workpiece. Therefore, the amount of particles embedded on the surfaces could not be determined accurately. The accuracy of the results, which depends on the area size examined under the SEM, is not reported. It is also not clear what “Three hundred and fifteen cylindrical samples.” as stated in the abstract mean. Findings do not contribute to any advancement in the field.
Reviewer 3 Report
To the author
In this manuscript, the impact of the type and hardness of materials on the number of embedded abrasive particles after airborne-particle abrasion is investigated. The topic is interesting; however, the investigation of background is not adequate and the method is unrefined (the method is old and/or how the study is novel is not explained). The reviewer considers that at this moment this manuscript does not merit publication. Regrettably, the reviewer believes reject is proper decision.